# Systematic review the efficacy and safety of cilostazol, pentoxifylline, beraprost in the treatment of intermittent claudication: A network meta-analysis

**Xinyu Liang** [1,2©¤‡] *, **Yuzhen Wang** [1©‡], **Cheng Zhao** [1©], **Yemin Cao** [1,2©] *

**1** Department of Peripheral Vascular, Shanghai TCM-Integrated Hospital Affiliated to Shanghai University of Traditional Chinese Medicine, Shanghai, China, **2** Clinical Faculty of Integrated Traditional Chinese and Western Medicine, Shanghai University of Traditional Chinese Medicine, Shanghai, China

© These authors contributed equally to this work.
¤ Current address: Shanghai University of Traditional Chinese Medicine, Pudong New Area, Shanghai, China
‡ These authors share first authorship on this work
* Professorcao2021@126.com (YC); 1064321689@qq.com (LX)

**Data Availability Statement:** All relevant data are within the manuscript and its Supporting Information files.

## Abstract

### Objective

To evaluate the efficacy and safety of cilostazol, pentoxifylline, beraprost for intermittent claudication due to lower extremity arterial occlusive disease.

### Methods

Randomized controlled clinical trials were identified from PubMed, Scopus, EMbase, Cochrane Library, Web of Science, China National Knowledge Infrastructure, SinoMed, Wanfang and Chongqing VIP databases, from the database inception to 31/12/2021. The outcome measures were walking distance measured by treadmill (maximum and pain-free walking distance), ankle-brachial index and adverse events. The quality of included studies was assessed by the Cochrane bias risk assessment tool. A network meta-analysis was carried out with Stata 16.0 software.

### Results

There were 29 RCTs included in the study, covering total 5352 patients. Cilostazol was ranked first for both maximum and pain-free walking distance, followed by beraprost and pentoxifylline. For cilostazol, pentoxifylline and beraprost, maximum walking distance increased by 62.93 95%CI(44.06, 81.79), 32.72 95%CI(13.51, 55.79) and 43.90 95%CI(2.10, 85.71) meters, respectively relative to placebo, and pain-free walking distance increased by 23.92 95%CI(11.24, 36.61), 15.16 95%CI(2.33, 27.99) and 19.78 95%CI(-3.07, 42.62) meters. For cilostazol, pentoxifylline, beraprost and cilostazol combined with beraprost, ankle-brachial index increased by 0.06 95%CI(0.04, 0.07), -0.01 95%CI(-0.08, 0.05), 0.18 95%CI(0.12, 0.23) and 0.23 95%CI(0.18, 0.27), respectively relative to placebo.

**Funding:** This study was supported by the National Natural Science Foundation of China [Project No. 82174382] and the Special Fund of Shanghai Science and Technology Innovation Action Plan [Project No. 20Z21900200] in the form of funds to YC.

**Competing interests:** The authors have declared that no competing interests exist.

The pentoxifylline and cilostazol was associated with a lower ratio of adverse events than beraprost and cilostazol combined with beraprost.

## Conclusion

Cilostazol, pentoxifylline and beraprost were all effective treatments for intermittent claudication; cilostazol with good tolerance was likely to be the most effective in walking distance, while beraprost and cilostazol combined with beraprost were more prominent in the ankle-brachial index.

## Introduction

Peripheral arterial disease (PAD), an atherosclerotic disease of the lower limbs, leads to shortage of blood flow and oxygen and nutrients to the lower extremities [1,2]. The common typical symptom of PAD is intermittent claudication (IC) that manifests as fatigue, pain, or spasms of lower extremity, and it is exhibited during mild exercise such as walking, but resolves after rest, resulting in restricted walking. According to recent studies, approximately 10–15‰ of people age >50 years have asymptomatic peripheral atherosclerotic disease, 5–10‰ have intermittent claudication symptoms [3–6]. Intermittent claudication not only reduces walking ability and quality of life, but also increases risk of serious complications such as major amputation and death [1,6]. With the aging population, the number will be projected to increase continuously, leading to be a heavy burden on the society and health care.

The treatment of IC involves the management of associated cardiovascular risk factors and improve walking symptoms, which can be addressed initially through some medical suggestions, such as supervised or unsupervised walking exercise, and lifestyle regulation (i.e., quit smoking and lose weight). Those are priorities for IC to relieve symptom, and when these are not effective, vasoactive drugs can be used commonly by vascular specialists to relieve walking symptom and to improve the quality of life [7]. These vasodilators may be administered for a long time, or until lower limb symptoms worsen and the patient requires surgical procedures (angioplasty, etc.). Although cilostazol, beraprost and pentoxifylline are usually applied in clinical practice, there is still a lack of evidence-based medical guidance comparing the efficacy of these drugs. A bayesian network meta-analysis allows multiple treatments to be simultaneously compared both direct and indirect evidence about therapeutic effects. Therefore, the study aims to systematically assess the efficacy of three vasoactive drugs for the treatment of IC related with PAD, with the purpose of providing comprehensive and reliable evidence in clinical practice by using a network meta-analysis.

## Methods

This study was performed in conformity to the Cochrane Handbook for the Systematic Review of Interventions and the Preferred Reporting Items for Systematic Review and Meta-Analyses (PRISMA) [8]. This project has been registered on PROSPERO CRD42022300419(https://www.crd.york.ac.uk/PROSPERO/#recordDetails). The PRISMA checklist was reported in S1 File.

### Inclusion and exclusion criteria

A systematic review was conducted to identify maximum walking distance (MWD), pain-free walking distance (PFWD), ankle-brachial index (ABI) and adverse event (AE) literature

concerning cilostazol, pentoxifylline and beraprost for the treatment of IC in people with PAD. MWD and PFWD were obtained by a treadmill test. The ankle-brachial index was obtained as the ratio of systolic blood pressures at the ankle to the systolic blood pressures of the upper extremity, which was a recognized method for detecting PAD by assessing the degree of lower limb ischemia. Adverse events were defined as patients who withdrew or dropped from the study due to adverse reaction, lower extremities and cardiovascular events (i.e., death, stroke, lower extremity surgery, or amputation at any level). To be eligible for inclusion, studies had to be randomized parallel controlled trials (RCTs), and they contained sufficient data to obtain effect sizes of interested outcome measures in the article. Study participants were intermittent claudication due to PAD, regardless of gender. Duplicate literatures and abstract-only studies were excluded. A minimum 12-week treatment period is considered necessary. Studies were excluded if they were editorials, opinion pieces, reviews, reports without full text where insufficient details were reported to allow inclusion, studies published not in Chinese or English.

## Literature search

We systematically looked for the following databases: the China National Knowledge Infrastructure (CNKI), SinoMed, WanFang Data, Chongqing VIP databases, PubMed, EMbase, and Scopus, Cochrane Library, Web of science. Additional records were searched for grey literature (unpublished studies) using the Chinese Clinical Trial Register and the ClinicalTrials. And some further trials were also hand-searched through industry submissions and relevant systematic reviews. A comprehensive search strategy was used, including beraprost, cilostazol, pentoxifylline, and arteriosclerosis obliterans, Peripheral arterial disease, PAD, ASO, and walking distance, walking time, maximum walking distance, pain-free walking distance, ankle-brachial index, MWD, PFWD, ABI.

## Data selection

Searched literatures were screened initially through title and abstract. The full texts of potential studies were got and further screened for eligibility based on preestablished inclusion and exclusion criteria. Two investigators (L.X.Y. and W.Y.Z.) independently scanned the potentially eligible literatures, extracted data, and cross-checked for each other, discussing openly or seeking a third opinion (C.Y.M.) when necessary. All excluded studies were marked with the reason for exclusion.

**Data extraction and quality evaluation.** Data were extracted with no blinding to authors or journal by one reviewer (W.Y.Z.) in a standard format and checked by another (Z.C). Information gained from the eligible studies included as follow: (1) the basic information of the study, including the title, first author, year, and journal; (2) study characteristics, including study location, treatments, doses, and duration; (3) relevant outcome measures in the study (such as MWD, PFWD, and ABI at baseline and at the end of the study, and adverse events); (4) additional information on the risk assessment of the study.

The quality of the including studies was assessed by one researcher (L.X.Y.) independently based on the Cochrane Risk of Bias Risk Assessment Tool recommended by the Cochrane Handbook version 5.3 [9,10]. The tool was commonly used to evaluate RCTs, mainly containing 7 items: random methods, allocation concealment, blinding researchers and participants, blinding outcome assessor, integrity of research data, selective reporting of research results, and other biases, and each entry classified into low risk, high risk, and unclear.

## Statistical analysis

The primary efficacy was analyzed using Stata16.0 and Review management 5.3 software to draw network diagrams and compare multiple interventions directly or indirectly. Statistical significance was defined as $P < 0.05$. For continuous outcome data, the analysis was performed using the treatment-specific data (sample means and standard difference) that were explicitly reported in the published studies. In some studies that did not report standard difference, the standard difference was derived using the reported mean and confidence interval for the difference between treatments in the geometric mean change from baseline, or mean range or standard error or by inverting the result of the test statistic. For categorical variable, the relative ratio (RR) was acquired by comparing the ratio of AE in the experimental group to the ratio of AE in the placebo group. The treatment effect of vasoactive drugs was ranked by the surface under cumulative ranking curve probabilities (SUCRA), and the SUCRA is expressed as a percentage, the larger the value, the better the efficacy. The consistency of results were tested by performing the node-splitting generalized linear mixed model to analysis the heterogeneity between studies. The model of consistency was fitted when the node split model was P value >0.05; otherwise, the inconsistency model was used. Heterogeneity for all pairwise comparisons was assessed by means of the Higgins' $I^2$ statistic, and $I^2 > 50\%$ was considered as statistically significant heterogeneity. The efficacy of drugs was evaluated by weighted mean difference (WMD, indicators changed from baseline), along with 95 percent confidence interval (CI). The safety of drugs was assessed by relative risk (RR) and its 95 percent confidence interval (CI). Robustness of conclusion was conducted by using the inverted funnel chart or assessing differences of clinical characteristics and methodologies between included studies.

**Results.** Using searching strategies, a total of 1206 articles was yielded after removing duplicates. Twenty-nine RCTs [11–39] involving 5352 patients with PAD were identified that met the inclusion criteria. Flowchart of research screening was shown in Fig 1. The basic information of qualified studies was shown in Table 1. The bias risk assessment of studies was shown in Figs 2 and 3. The evidence base formed a network of studies comparing two or three medications, as shown in Fig 4. In addition to two trials being direct comparison of cilostazol and pentoxifylline, one trial being a comparison with cilostazol and cilostazol together with beraprost, twenty-six of the included researches were placebo-controlled trials, with eleven comparing with cilostazol, eight being a comparison with pentoxifylline, five comparing with beraprost, and three being a three-arm comparison of cilostazol and pentoxifylline. Thirteen of the clinical trials were performed in USA, two in France, one in Sweden, five in China, one in India, one in Brazil, one in Poland, one in Italy, and four in England. The results of Higgins 'I^2 statistic indicated that all pairwise comparisons showed varying degrees of heterogeneity, as shown in S1 Table.

## Network meta-analysis

Most researches were two-arm placebo-controlled trials being a comparison either cilostazol, pentoxifylline or beraprost with placebo (Fig 4). In addition, the network model consisted of a closed loop, which takes into assessing the consistency between the direct and indirect evidence about the efficacy of cilostazol and pentoxifylline. The beraprost lacked a loop in the network evidence, which means there is consistency on the efficacy evaluation.

**Maximum walking distance.** A total of 22 studies [11–14,17–28,30–35] reported MWD, involving 4174 patients included. The network evidence was shown in Fig 4. The result of inconsistency model showed that $P = 0.84 > 0.05$, suggesting that the consistency model was fitly applied for the analysis. The values indicated the weighted mean difference and 95% CI of the medicines in row compared with the drugs in column. Drugs had been sorted according to

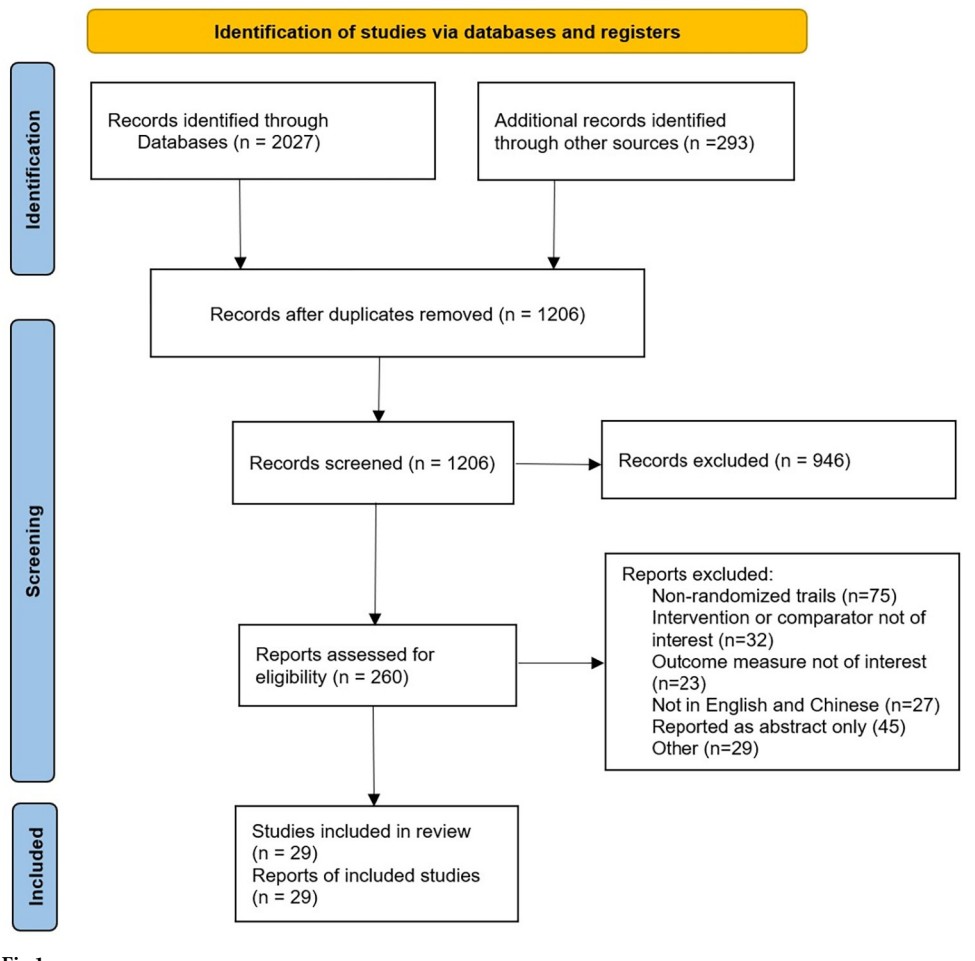

**Fig 1.**

the mean rank. Bold figures indicated difference was statistically significant. All drugs were linked with an increase in MWD compare to placebo. Cilostazol had the greatest effect on MWD with an increase of 62.93 meters (95 percent credible interval (CI) 44.06 to 81.79, $I^2$ = 49.4%, P < 0.05), compared with placebo, and for pentoxifylline and beraprost, MWD increased by 32.72 meters (95%CI 12.97 to 52.46, $I^2$ = 76.0%, P < 0.05) and 43.90 meters (95% CI 2.10 to 85.71, $I^2$ = 86.7%, P < 0.05), respectively, as shown in Table 2. The SUCRA probabilities and the mean rank showed that cilostazol was ranked first in improving MWD, followed by beraprost, pentoxifylline and placebo, as shown in Tables 2 and 6. The rank probabilities of second, third and subsequent ranks for each intervention were shown in S2–S5 Tables.

**Pain-free walking distance.** A total of 18 studies [11–13,15,16,18,20,22–28,30,32,33,35] reported the results of PFWD, covering 3538 patients. The network evidence was shown in Fig 4. The result of inconsistency model test showed that P = 0.16 > 0.05, meaning that the consistency model was used for the network meta-analysis. Relative to placebo, three vasodilators could increase PFWD, although there was some uncertainty about the efficacy of beraprost: 19.78 meters (95 percent credible interval (CI) −3.07 to 42.62, $I^2$ = 62.7%, P > 0.05). Compared with placebo, cilostazol had best effect on PFWD with an increase of 23.92 meters (95%CI 11.24 to 36.61, $I^2$ = 42.7%, P < 0.05), while pentoxifylline increased by 15.16 meters (95%CI 2.33 to 27.99, $I^2$ = 0.0%, P < 0.05), as shown in Table 3. The SUCRA probabilities and the

**Table 1. Characteristics of included studies.**

| Authors & years | country | patients | male | age | number | groups | outcome |
|---|---|---|---|---|---|---|---|
| Brass 2012 [18] | USA | IC | 70 | 44.0–82.0 | 87 | placebo | ①②④ |
| | | | 70 | 50.0–84.0 | 89 | cilostazol | |
| Huisinga 2010 [19] | USA | IC | NA | 66.4±10.1 | 11 | pentoxifylline | ② |
| | | | | 67.0±7.4 | 9 | cilostazol | |
| Donnel 2009a [26] | Kingdom | IC, Non-DM | 26 | NA | 39 | cilostazol | ①②③④ |
| | | | 27 | | 41 | placebo | |
| Donnell 2009b [25] | Kingdom | IC | 34 | 64.2 | 51 | cilostazol | ①②④ |
| | | | 39 | 66.1 | 55 | placebo | |
| Donnell 2009c [27] | Kingdom | IC, DM | NA | NA | 12 | cilostazol | ①②④ |
| | | | | | 14 | placebo | |
| Singh 2009 [35] | India | IC | NA | NA | 26 | pentoxifylline | ①② |
| | | | | | 28 | cilostazol | |
| | | | | | 25 | placebo | |
| Creager 2008 [28] | USA | IC | 67 | 67.2 | 86 | pentoxifylline | ①②④ |
| | | | 69 | 66.7 | 84 | placebo | |
| de Albuquerque 2008 [14] | Brazil | IC | NA | 64.0 ± 10 | 6 | pentoxifylline | ② |
| | | | | 64.0 ± 9.0 | 12 | cilostazol | |
| Krauss 2007 [20] | Poland | IC | NA | 65.9 | 20 | pentoxifylline | ①②③ |
| | | | | 65.9 | 20 | placebo | |
| Mohler 2003 [30] | USA | IC | 306 | 65.9 | 385 | beraprost | ①②④ |
| | | | 279 | 65.7 | 377 | placebo | |
| Strandness 2002 [34] | USA | IC | 102 | 63.1±10.2 | 133 | cilostazol | ②④ |
| | | | 100 | 64.4±10.2 | 129 | placebo | |
| Lee 2001 [21] | China | IC | 13 | 66.0±9.0 | 16 | cilostazol | ②④ |
| | | | 13 | 68.0±5.0 | 16 | pentoxifylline | |
| | | | 14 | 69.0±6.0 | 16 | placebo | |
| Mohler 2001 [29] | USA | IC | 231 | 63.8±9.3 | 308 | cilostazol | ③ |
| | | | 230 | 64.8±9.7 | 299 | placebo | |
| Dawson 2000 [12] | USA | IC | 172 | 66.0±9.0 | 227 | cilostazol | ①②④ |
| | | | 181 | 66.0±9.0 | 232 | pentoxifylline | |
| | | | 176 | 66.0±9.0 | 239 | placebo | |
| Lièvre 2000 [23] | France | IC | 177 | NA | 209 | beraprost | ①②④ |
| | | | 179 | | 213 | placebo | |
| Beebe 1999 [11] | USA | IC | 130 | 64.3 ± 8.5 | 175 | cilostazol | ①②④ |
| | | | 131 | 65.1 ± 9.3 | 170 | placebo | |
| Dawson 1998 [13] | USA | IC | 38 | 66.0 ± 1.1 | 54 | cilostazol | ①② |
| | | | 24 | 67.0 ± 2.0 | 27 | placebo | |
| Elam 1998 [17] | USA | IC | 83 | 66.7 | 95 | cilostazol | ②③④ |
| | | | 76 | 65.8 | 94 | placebo | |
| Money 1998 [31] | USA | IC | 90 | 64.8 ± 9.4 | 119 | cilostazol | ②③④ |
| | | | 90 | 64.5 ± 8.8 | 120 | placebo | |
| Lievre 1996 [22] | France | IC | 83 | 62.0±10.0 | 42 | beraprost | ①②④ |
| | | | 80 | 61.0±11.0 | 41 | placebo | |
| Dettori 1989 [15] | Italy | IC | 33 | 62.0±5.0 | 37 | pentoxifylline | ①③④ |
| | | | 35 | 59.0±8.0 | 37 | placebo | |
| Lindgärde 1989 [24] | Sweden | IC | 63 | 65.0±7.0 | 76 | pentoxifylline | ①②④ |
| | | | 58 | 64.0±8.0 | 74 | placebo | |

*(Continued)*

**Table 1.** (Continued)

| Authors & years | country | patients | male | age | number | groups | outcome |
|---|---|---|---|---|---|---|---|
| Reich 1987 [33] | USA | IC | 18 | 48.0–71.0 | 21 | pentoxifylline | ①② |
| | | | 15 | 49.0–70.0 | 17 | placebo | |
| Donaldson 1984 [16] | England | IC | 31 | 37.0–75.0 | 40 | pentoxifylline | ①③④ |
| | | | 31 | 37.0–76.0 | 40 | placebo | |
| Porter 1982 [32] | USA | IC | NA | NA | 67 | pentoxifylline | ①②④ |
| | | | | | 61 | placebo | |
| Liu 2015 [36] | China | IC, DM | 16 | 67.1±9.3 | 43 | cilostazol | ③④ |
| | | | 14 | 65.6±7.8 | 44 | placebo | |
| Zhang 2011 [37] | China | IC, DM | 14 | 69.5±11.2 | 24 | beraprost | ③④ |
| | | | 14 | 65.0±9.6 | 22 | placebo | |
| Hu 2017 [38] | China | IC, DM | 23 | 64. 8 | 46 | beraprost | ③④ |
| | | | 20 | 65. 0 | 41 | placebo | |
| Li 2013 [39] | China | IC, DM | NA | NA | 24 | B + C | ③④ |
| | | | | | 24 | cilostazol | |

[a]Table footnotes: IC: Intermittent claudication; DM: Diabetic patients; B+C: Beraprost combined with cilostazol; NA: Not report; ①: MWD (maximum walking distance); ②:PFWD (pain-free walking distance); ③:ABI (ankle-brachial index); ④:AE (adverse events).

mean rank showed that cilostazol was ranked first in the improvement of PFWD, followed by beraprost, pentoxifylline and placebo, as shown in Tables 3 and 6.

**Ankle-brachial index.** There was a total of 11 studies [15–17,20,26,29,31,36–39] reporting ABI, including 1577 patients. The network evidence was shown in Fig 4. Since the included studies did not form a loop, no consistency test was conducted. Cilostazol and beraprost could increase in terms of ABI relative to placebo ($I^2 > 50\%$, $P < 0.05$), while there was some uncertainty about the efficacy of pentoxifylline: -0.01 (95% CI −0.08 to 0.05, $I^2 = 0.0\%$, $P > 0.05$). Table 4 showed the different efficacy of cilostazol, pentoxifylline, beraprost in improving ABI, sorted by the mean rank. Meta-analysis results showed that SUCRA probabilities was beraprost combined with cilostazol > beraprost > cilostazol > placebo> pentoxifylline, as shown in Tables 4 and 6.

**Adverse events.** There were 23 studies [11,12,15–18,21–28,30–32,34,36–39] reporting AE as an outcome of interest, including 4346 patients. The network evidence relationship was

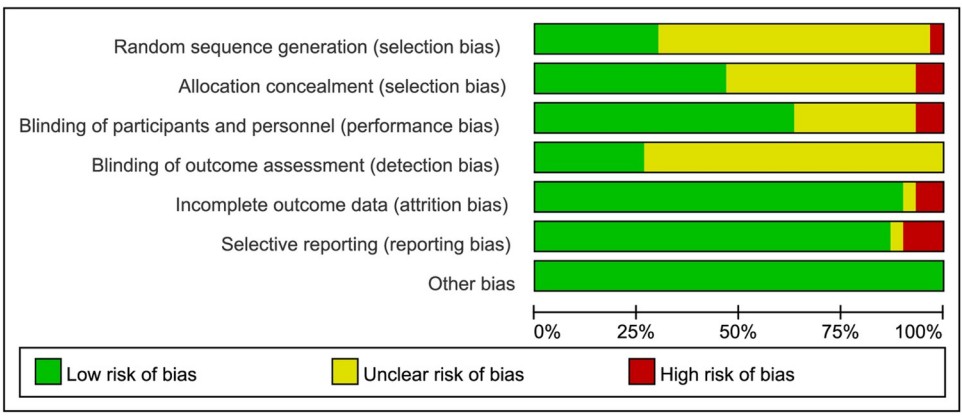

**Fig 2.**

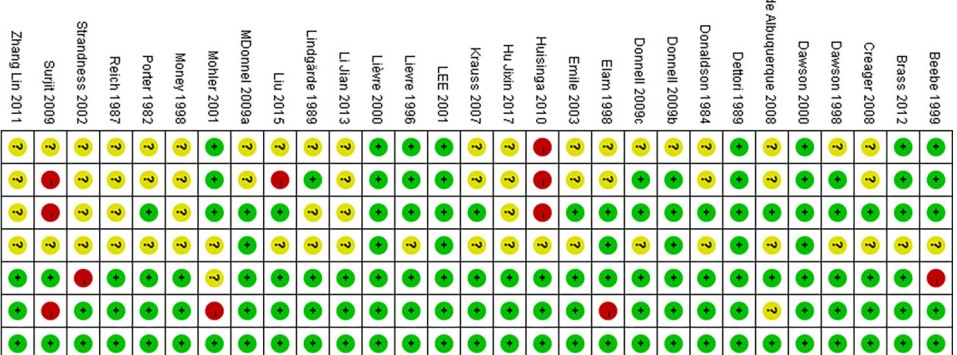

Fig 3.

shown in Fig 4. The result of inconsistency model test showed that P = 0.35 > 0.05, suggesting that the consistency model was fitted for the analysis. All drugs had adverse reactions of varying degrees, and Table 5 shows the relative risk of different drugs use in AE, sorted by the mean rank. Network meta-analysis results showed that the SUCRA probabilities was placebo > pentoxifylline > cilostazol > beraprost combined with cilostazol > beraprost, as shown in Tables 5 and 6.

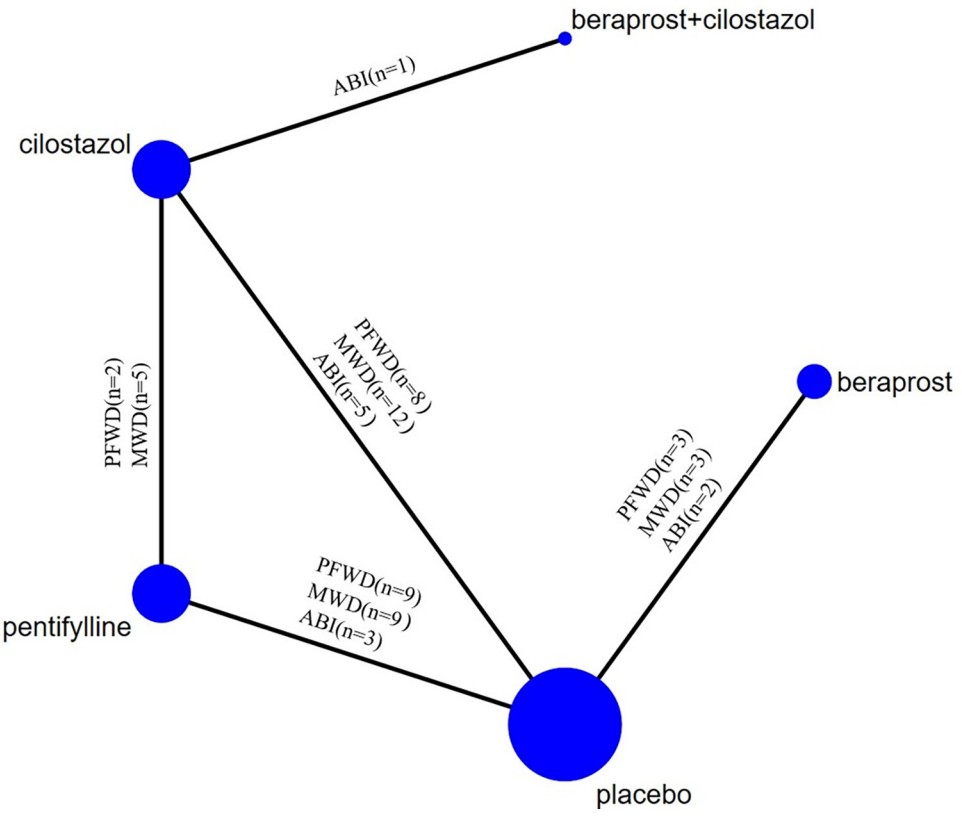

Fig 4.

**Table 2. The efficacy of vasoactive drugs in MWD (meter) and their 95 percent confidence intervals.**

| Mean Rank | drug | placebo | pentoxifylline | beraprost |
|---|---|---|---|---|
| 1.2 | cilostazol | **62.93 (44.06,81.79)** | **28.28 (4.52,52.04)** | 19.03 (-26.89,64.95) |
| 2.2 | beraprost | **43.90 (2.10,85.71)** | 9.25 (-37.67,56.17) | |
| 2.6 | pentoxifylline | **32.72 (12.97,52.46)** | | |
| 4 | placebo | | | |

[a]Table footnotes: The values indicated the weighted mean difference and 95% CI of the medicines in row compared with the drugs in column; Bold numbers mean the difference was statistically significant (P<0.05).

**Table 3. The efficacy of vasoactive drugs in PFWD (meter) and their 95 percent confidence intervals.**

| Mean Rank | Drug | placebo | pentoxifylline | beraprost |
|---|---|---|---|---|
| 1.5 | cilostazol | **23.92 (11.24,36.61)** | 8.76 (-7.46,24.98) | 4.15 (-22.11,30.41) |
| 2.1 | beraprost | 19.78 (-3.07,42.62) | 4.61 (-21.79,31.02) | |
| 2.4 | pentoxifylline | **15.16 (2.33,27.99)** | | |
| 3.9 | placebo | | | |

[a]Table footnotes: The values indicated the weighted mean difference and 95% CI of the medicines in row compared with the drugs in column; Bold numbers mean the difference was statistically significant (P<0.05).

**Table 4. The efficacy of vasoactive drugs in improving ABI and their 95 percent confidence intervals.**

| Mean Rank | Drug | placebo | pentoxifylline | cilostazol | beraprost |
|---|---|---|---|---|---|
| 1.1 | B+C | **0.23 (0.18,0.27)** | **0.24 (0.17,0.32)** | **0.17 (0.13,0.21)** | 0.05 (-0.02,0.12) |
| 1.9 | beraprost | **0.18 (0.12,0.23)** | **0.19 (0.11,0.27)** | **0.12 (0.07,0.18)** | |
| 3 | cilostazol | **0.06 (0.04,0.07)** | **0.07 (0.01,0.13)** | | |
| 4.7 | pentoxifylline | -0.01 (-0.08,0.05) | | | |
| 4.3 | placebo | | | | |

[a]Table footnotes: B+C was referred to beraprost combined with cilostazol; The values indicated the weighted mean difference and 95% CI of the medicines in row compared with the drugs in column; Bold numbers mean the difference was statistically significant (P<0.05).

**Table 5. The relative risk of vasoactive drugs in AE and their 95 percent confidence intervals.**

| Mean Rank | Drug | beraprost | BC | cilostazol | pentoxifylline |
|---|---|---|---|---|---|
| 1.3 | placebo | **0.41 (0.28,0.61)** | 0.50 (0.09,2.66) | **0.69 (0.49,0.98)** | 0.70 (0.46,1.06) |
| 2.8 | pentoxifylline | 0.59 (0.33,1.04) | 0.71 (0.13,3.90) | 0.99 (0.63,1.56) | |
| 2.9 | cilostazol | 0.59 (0.35,1.01) | 0.71 (0.14,3.70) | | |
| 3.5 | B+C | 0.83 (0.15,4.68) | | | |
| 4.5 | beraprost | | | | |

[a]Table footnotes: B+C was referred to beraprost combined with cilostazol; The values indicated the relative risk and 95% CI of the medicines in row compared with the drugs in column; Bold numbers mean the difference was statistically significant (P<0.05).

**Table 6. The surface under cumulative ranking curve probabilities (SUCRA) for outcomes.**

| drugs | PFWD(%) | MWD(%) | ABI(%) | AE(%) |
|---|---|---|---|---|
| placebo | 1.7 | 0.8 | 16.6 | 92.9 |
| cilostazol | 82.8 | 92.8 | 49.7 | 53.3 |
| pentoxifylline | 52.2 | 45.6 | 8.8 | 54.1 |
| beraprost | 63.3 | 60.8 | 77.1 | 11.8 |
| B+C | NA | NA | 97.9 | 38 |

[a]Table footnotes: NA: Not report; B+C was referred to beraprost combined with cilostazol.

**Comprehensive evaluation.**  Basing on SUCRA of efficacy and safety, a comprehensive evaluation of all treatments was made, suggesting that cilostazol had a best effect on improving walking distance, while cilostazol combined with beraprost have a best effect on ABI, as showed in Fig 5.

**Robustness of conclusion.**  The Reich [31] study reported PFWD and MWD but no specific treadmill protocol provided, giving rise to a potential affect the results of this study. To evaluate the robustness of the results of MWD and PFWD in our study, a sensitivity analysis was performed excluding Reich's study from the network meta-analysis. After excluding this study, the improvement in PFWD increased from 15.16 (95%CI 2.33 to 27.99) meters to 15.83 (95%CI 2.00 to 29.67) meters, while the improvement in MWD increased from 32.72 (95%CI 12.97 to 52.46) meters increased to 34.65 (95% CI 13.51 to 55.79) meters. The conclusions of pentoxifylline in the improvement of PFWD and MWD were obviously unimpacted including data from Reich study.

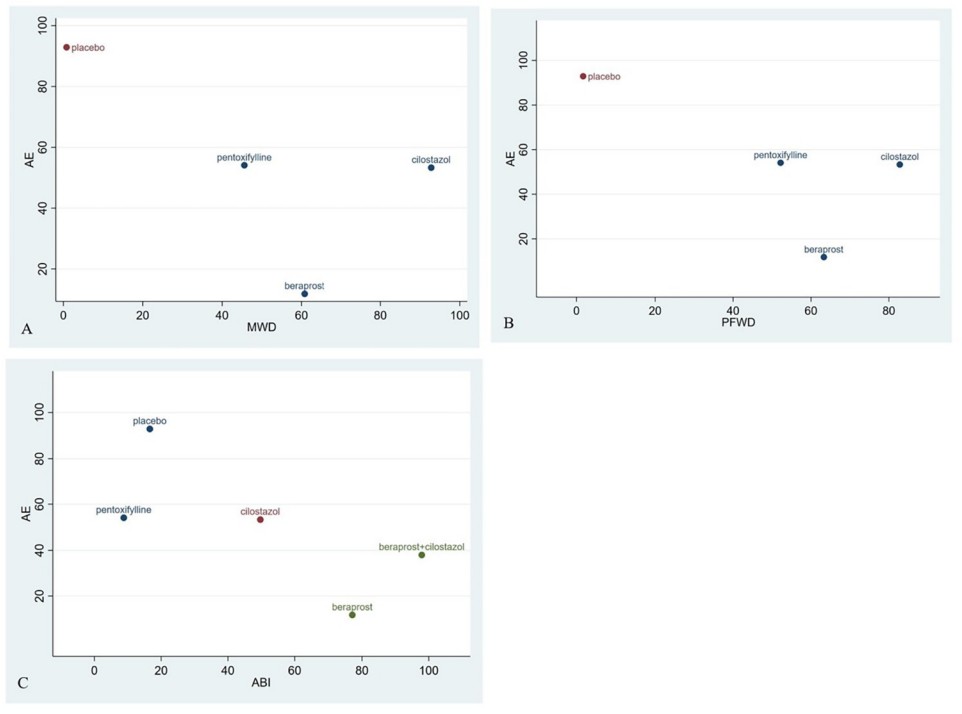

**Fig 5.**

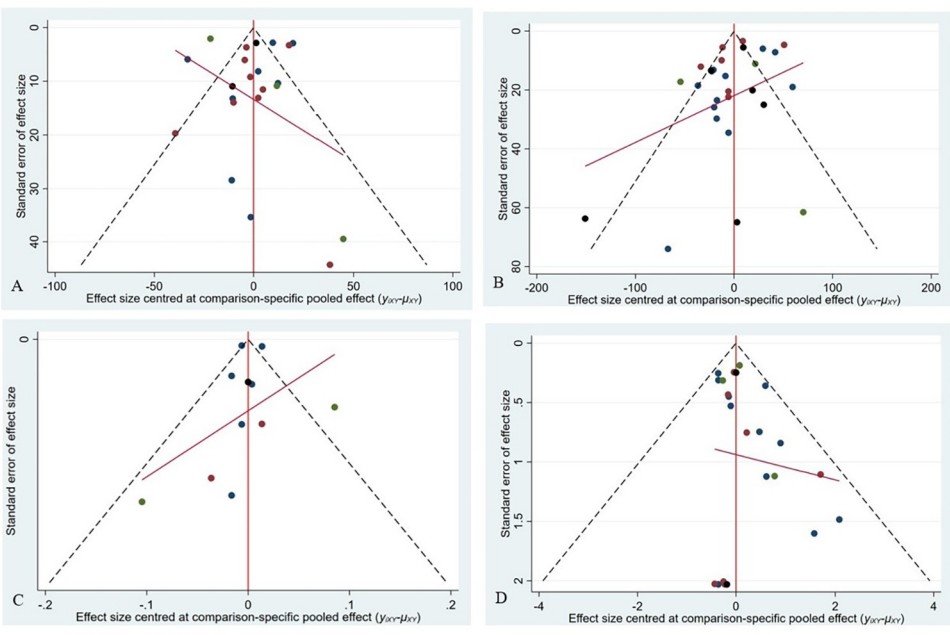

**Fig 6.**

The inverted funnel chart was made with the PFWD, MWD, ABI and AE, as shown in Fig 6. The MWD and PFWD was basically symmetrically distributed, suggesting that the publication bias was small (Fig 6A and 6B). The ABI and AE were generally scattered and slightly biased, indicating there may be a certain publication bias (Fig 6C and 6D).

## Discussion

In the clinical treatment of IC, patients usually been administrated vasoactive drugs to increase walking distance, in addition to walking exercise and the management of risk factors (i.e., controlling lipids, blood glucose and blood pressure). Beraprost has several therapeutic effects, including protecting vascular endothelial, inhibiting platelet aggregation and reducing inflammation, and can improve ABI, walking distance and feeling of cold [40]. Cilostazol, a phosphodiesterase 3 inhibitor with antiplatelet aggregation and vasodilation effects, is used as a treatment to improve walk symptoms in IC patients with PAD [41]. Pentoxifylline, a vasoactive drug, has been authorized for the medical treatment of individuals with IC, which decreases blood viscosity, improves erythrocyte flexibility, promotes microcirculatory flow and increases tissue oxygen concentration [41]. Vasoactive drugs (i.e., cilostazol and beraprost) are applied when symptoms of IC persist and affect quality of life.

The studies by Broderick C et al. [42] and Brown T et al. [43] had shown that cilostazol and pentoxifylline might be effective drugs to improve walking distance. Ma Bo and co-workers [44] recently published a meta-analysis of five medications (beraprost, aspirin, etc.) that included twenty-seven trials are not part of the present review (7 studies evaluated walking distance, 14 were studies less than 12 weeks). The conclusion from Ma Bo et al. [44] was that beraprost had better efficacy in improving walking distance, but the confidence interval of the result was very large [516.87 meters, 95%CI (-1205.36, 2239.10)]. Therefore, it was necessary to further synthetically compare between vasoactive drugs, providing guidance for clinicians in practice.

Our study contemporaneously evaluated the therapeutic effects of cilostazol, beraprost and pentoxifylline for the treatment of IC due to PAD. Compared with placebo, cilostazol and pentoxifylline could significantly increase walking distance (P<0.05), but there was some uncertainty about the efficacy of beraprost (P>0.05). Cilostazol were ranked top 1 in MWD and PFWD among IC patients, which was followed by beraprost. Vasodilators except pentoxifylline significantly improve ABI compared to placebo (P<0.05). Cilostazol combined with beraprost was ranked top 1 in improving ABI, which was followed by beraprost and cilostazol. Although, compared with placebo, the improvement of beraprost was three times that of cilostazol [0.18 VS 0.06, P<0.05], this ABI increment of the former did not appear to bring any benefit in walking distance. Cilostazol combined with beraprost greatly increased ABI, but it had poor tolerance and lacked an assessment of walking distance. In addition, pentoxifylline was not significantly superior to placebo in the improvement of PFWD and ABI, which might be due to the small sample of studies.

The main adverse reactions of cilostazol are headache, abnormal stools and dizziness, and these were usually mild and transient [45]. The common complaint of pentoxifylline is gastrointestinal symptoms, occurring in lower 3% of patients [46]. Although the incidence of beraprost is generally under 1.2% for each symptom including headaches, hectic fever, diarrhea and nausea [47], the risk and severity of AE are higher than cilostazol and pentoxifylline. PAD featured with arterial occlusion of the lower extremity is a type of systemic arterial diseases. Adverse cardiovascular events are common in patients with IC due to PAD. However, the vasodilators compared with placebo did not appear to reduce serious cardiovascular adverse events (i.e., myocardial ischemia, stroke and death). Perhaps the relatively short follow-up time (12–24 weeks) in clinical trials may not be enough to draw definitive results.

There were some limitations. First, some RCTs included studies did not report randomization, assignment hiding method and blind method, which may be selection and measurement bias. Second, Investigators did not follow a common protocol of treadmill test to assess PFWD and MWD, which had brought about varying degrees of heterogeneity between studies in the estimate of effect size. In addition, the findings still required to be further proved by large sample and high-quality clinical studies due to limited data on direct comparison of different vasodilators (Only 3 studies directly compared cilostazol with pentoxifylline). In addition to exploring the direct comparison of the efficacy of different vasodilators, it is also crucial for future research to evaluate the efficacy of vasodilators combined with other drugs (aspirin, atorvastatin, etc.) in the treatment of IC patients.

## Conclusion

Our study suggested that cilostazol might be ideal vasodilator in terms of walking distance and safety for the treatment of IC due to PAD, while beraprost combined with cilostazol have a better effect on ABI. Although we provided evidence for ranking the therapeutic efficacy of vasoactive medications, there are some limitations in the study. Future high-quality RCTs should be performed to fully verify the different efficacy between drugs for a better clinical practice.

## Supporting information

**S1 Table. Heterogeneity assessment of all pairwise comparisons for different outcomes.**
(DOCX)

**S2 Table. The ranking probabilities in MWD.**
(DOCX)

**S3 Table. The ranking probabilities in PFWD.**
(DOCX)

**S4 Table. The ranking probabilities in ABI.**
(DOCX)

**S5 Table. The ranking probabilities in AE.**
(DOCX)

**S1 File. PRISMA_2020_checklist.**
(DOCX)

## Author Contributions

**Conceptualization:** Xinyu Liang, Yemin Cao.

**Data curation:** Xinyu Liang, Yuzhen Wang, Cheng Zhao, Yemin Cao.

**Formal analysis:** Xinyu Liang, Yuzhen Wang, Cheng Zhao, Yemin Cao.

**Investigation:** Yuzhen Wang, Cheng Zhao.

**Supervision:** Yemin Cao.

**Visualization:** Cheng Zhao, Yemin Cao.

**Writing – original draft:** Xinyu Liang.

**Writing – review & editing:** Xinyu Liang, Yemin Cao.

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
