## [Decision Letter · Decision Letter 0]

10 Jul 2022

PONE-D-22-10401Systematic review the efficacy and safety of cilostazol, pentoxifylline, berprost in the treatment of intermittent claudication: a network meta-analysisPLOS ONE

Dear Dr. liang,

Thank you for submitting your manuscript to PLOS ONE. After careful consideration, we feel that it has merit but does not fully meet PLOS ONE’s publication criteria as it currently stands. Therefore, we invite you to submit a revised version of the manuscript that addresses the points raised during the review process.

We look forward to receiving your revised manuscript.

Kind regards,

Tariq Jamal Siddiqi

Academic Editor

PLOS ONE

Journal Requirements:

Reviewers' comments:

Reviewer's Responses to Questions

**Comments to the Author**

1. Is the manuscript technically sound, and do the data support the conclusions?

Reviewer #2: Yes

2. Has the statistical analysis been performed appropriately and rigorously? 

Reviewer #2: Yes

3. Have the authors made all data underlying the findings in their manuscript fully available?

Reviewer #2: Yes

4. Is the manuscript presented in an intelligible fashion and written in standard English?

Reviewer #2: Yes

5. Review Comments to the Author

Reviewer #2: Liang et al. conducted a study on “Systematic review the efficacy and safety of cilostazol, pentoxifylline, beraprost in the treatment of intermittent claudication: a network meta-analysis”, in which they explored the efficacy and safety of cilostazol, pentoxifylline, and beraprost for the treatment of intermettient claudication as a result of lower extremity arterial occlusive disease. They found that cilostazol presented with the greatest improvement in both maximum and pain-free walk distance outcomes. A combination therapy of beraprost and cilostazol presented with the greatest improvement in ankle-brachial index outcome and of the active therapies, pentoxifylline, followed closely by cilostazol, presented with the lowest rates of adverse events.

In my opinion, this study may be improved by incorporating the following edits:

1. The novelty of this study needs to be highlighted more clearly. Do previous meta-analyses or network meta-analyses exist for these outcomes? If so, what unique characteristic of this study distinguishes it from previous studies conducted to this end? If previous studies exist, the authors should refer to them and clearly identify distinguishing features in their study which they believe would meaningfully improve the accuracy and reliability of their results compared with their predecessors.

2. The authors frequently misspell “beraprost” to “berprost”. As it is one of the main therapies under consideration, the authors should take care to correct this error in all instances, especially in the title of the study.

3. The authors should mention the 95% confidence interval values for the outcomes in the abstract.

4. In line 35, please rephrase the sentence to “Cilostazol, pentoxifylline and berprost are all effective treatments for intermittent claudication”.

5. In the methodology section, please specifically highlight whether the PRISMA-NMA or other guidelines were referred to while performing this network meta-analysis.

6. The authors should evaluate heterogeneity for all pairwise comparisons using Higgins' I2 statistic.

7. In the “Data Selection” portion of the methodology section, the initials of the authors conducting the literature search and the author consulted for a third opinion in case of disagreements should be highlighted.

8. In the “Data extraction and Quality Evaluation” portion of the methodology section, the initials of the author conducting the data extraction and the author checking the extraction should be highlighted.

9. Please specifically clarify if the quality assessment of the included studies was conducted by a single author or if it was conducted independently by multiple authors who resolved any differences of opinion by consulting a third author. Initials of all authors involved in the process should be highlighted as well.

10. In line 119, the name of the statistical software, in this case Stata, should be provided alongside the version number. Please change the sentence to include the name of the software.

11. Please present p-values for the outcomes evaluated in the study, in both the results section and in the attached tables in order to highlight significance of results. Please also mention in the methodology section that a p-value of p<0.05 was considered significant in all cases.

12. Please briefly justify the cutoff value of 40 years old for this study. Please provide an adequate explanation as to why this is the only appropriate age group in which to explore this outcome or specifically clarify that this study is limited only to older populations.

13. The authors appear to be utilizing ranking probability based solely on the probability of a certain intervention being ranked as the best. As mentioned in the Cochrane Handbook, inference on this basis is considered misleading and should be avoided. To this end, the authors should amend the analysis by accounting for rank probabilities of second, third and subsequent ranks for each intervention and utilising ranking measures such as the mean ranks, median ranks or cumulative ranking probabilities, as appropriate, to arrive at a definitive conclusion. Alternatively, the authors could mention this flaw in the methodology in the limitations portion of their discussion with the disclaimer that as only the probabilities of highest rank were accounted for, drawing conclusions purely on the basis of these findings may be erroneous.

14. If the authors decide to account for other ranking probabilities, please present the ranking data alongside the subsequent ranking measures to this end, as a separate table. If the authors decide against this, it would be appreciated if the authors could indicate the current rankings in Table 6 for each outcome and intervention therein.

15. In the results section, lines 155-157, please consider removing the sentence “However, a bayesian network meta-analysis allows multiple treatments to be simultaneously compared both direct and indirect evidence about therapeutic effects.” This sentence is better suited for the introduction or discussion sections.

16. In Table 1, the descriptions of 1, 2, 3 and 4, as presented in the outcome column, and which specific outcome they each refer to, should be presented in the note associated with the table.

17. In lines 223-224 please provide 95% confidence interval values for the change in outcome measures generated as a result of the sensitivity analysis.

18. Please clarify where the value of 32.72 in line 224 was obtained from. The value presented in Table 2 to this end appears to be 34.65 (95% CI 13.51 to 55.79).

19. The second paragraph of the discussion section, specifically from line 247 to 266, requires significant improvement. The grammar and sentence structure makes the interpretation of results therein extremely confusing. This section should be revised and structured more carefully to prevent confusion.

20. Lines 251 to 255 also appear to make baseless conclusions such as “Compared with pentoxifylline and berprost, cilostazol could increase by 8.76 and 4.15 meters, respectively”. As stated by the authors, the analysis to this end presented with statistical non-significance between outcomes, which directly contradicts this statement. While a hierarchy may be proposed on the basis of ranking probability, statements such as the above, made despite a clear indication of statistical non-significance in the outcome measure, are still considered erroneous.

21. Please highlight the avenues for future research which have been opened up as a result of this study, or are needed to contextualize and further the findings in this study, more exhaustively. This could be in the form of studies evaluating more direct comparisons to complete more closed loops for future network meta-analyses to this end or studies investigating the efficacy of cilastozol plus beraprost in the walk distance outcome measures.

6. PLOS authors have the option to publish the peer review history of their article (what does this mean?). If published, this will include your full peer review and any attached files.

Reviewer #2: **Yes: **Muhammad Talha Maniya

---

## [Author Response · Author response to Decision Letter 0]

26 Jul 2022

1. The novelty of this study needs to be highlighted more clearly. Do previous meta-analyses or network meta-analyses exist for these outcomes? If so, what unique characteristic of this study distinguishes it from previous studies conducted to this end? If previous studies exist, the authors should refer to them and clearly identify distinguishing features in their study which they believe would meaningfully improve the accuracy and reliability of their results compared with their predecessors.

Response: We added relevant statements about the innovative nature of this studyin the second paragraph of the Discussion section.

 2. The authors frequently misspell “beraprost” to “berprost”. As it is one of the main therapies under consideration, the authors should take care to correct this error in all instances, especially in the title of the study.

Response: The misspelled word had been corrected in the manuscript.

3. The authors should mention the 95% confidence interval values for the outcomes in the abstract.

Response: The summary had been supplemented with 95% confidence interval for the outcomes.

4. In line 35, please rephrase the sentence to “Cilostazol, pentoxifylline and berprost are all effective treatments for intermittent claudication”.

Response: We had corrected the sentence in the abstract of line 35.

5. In the methodology section, please specifically highlight whether the PRISMA-NMA or other guidelines were referred to while performing this network meta-analysis.

Response: We had added this in the Methodology section.

6. The authors should evaluate heterogeneity for all pairwise comparisons using Higgins' I2 statistic.

Response: We had added statements about heterogeneity assessment in the statistical analysis section. And the results of heterogeneity for all pairwise comparisons were supplemented in S1 Table.

7. In the “Data Selection” portion of the methodology section, the initials of the authors conducting the literature search and the author consulted for a third opinion in case of disagreements should be highlighted.

Response: We have added the initials of the authors (L.X.Y. and W.Y.Z.) conducting the literature search and the author (C.Y.M.) consulted for a third opinion in the “Data Selection” portion of the methodology section.

8. In the “Data extraction and Quality Evaluation” portion of the methodology section, the initials of the author conducting the data extraction and the author checking the extraction should be highlighted.

Response: We have added the author (W.Y.Z.) conducting the data extraction and the author (Z.C.) checking the extraction in the “Data extraction and Quality Evaluation” portion of the methodology section.

9. Please specifically clarify if the quality assessment of the included studies was conducted by a single author or if it was conducted independently by multiple authors who resolved any differences of opinion by consulting a third author. Initials of all authors involved in the process should be highlighted as well. 

Response: We have added the author (Z.C.) conducting the quality assessment of the included studies in the “Data extraction and Quality Evaluation” portion of the methodology section.

10. In line 119, the name of the statistical software, in this case Stata, should be provided alongside the version number. Please change the sentence to include the name of the software.

Response: We have supplemented the name of the software and the version number.

11. Please present p-values for the outcomes evaluated in the study, in both the results section and in the attached tables in order to highlight significance of results. Please also mention in the methodology section that a p-value of p<0.05 was considered significant in all cases.

Response: We have supplemented p-values of the main outcomes in the results section and in the attached tables. And “Statistical significance was defined as P < 0.05” was added in the methodology section.

12. Please briefly justify the cutoff value of 40 years old for this study. Please provide an adequate explanation as to why this is the only appropriate age group in which to explore this outcome or specifically clarify that this study is limited only to older populations.

Response: Peripheral arterial disease is most common in people over 40 years old, and we did not exclude any studies for this reason. We decide to delete this description for better understanding. 

13. The authors appear to be utilizing ranking probability based solely on the probability of a certain intervention being ranked as the best. As mentioned in the Cochrane Handbook, inference on this basis is considered misleading and should be avoided. To this end, the authors should amend the analysis by accounting for rank probabilities of second, third and subsequent ranks for each intervention and utilising ranking measures such as the mean ranks, median ranks or cumulative ranking probabilities, as appropriate, to arrive at a definitive conclusion. Alternatively, the authors could mention this flaw in the methodology in the limitations portion of their discussion with the disclaimer that as only the probabilities of highest rank were accounted for, drawing conclusions purely on the basis of these findings may be erroneous.

Response: To draw a definitive conclusion, we decided to utilize SUCRA probabilities and mean rank in the analysis, but the ranking of drugs did not change.

14. If the authors decide to account for other ranking probabilities, please present the ranking data alongside the subsequent ranking measures to this end, as a separate table. If the authors decide against this, it would be appreciated if the authors could indicate the current rankings in Table 6 for each outcome and intervention therein.

Response: We added the mean rank to the analysis of interested outcomes (MWD, PFWD, ABI and AE) in the table 2 – 5. In addition，we also added the rank probabilities of second, third and subsequent ranks for each intervention to supplementary materials (S2 Table).

15. In the results section, lines 155-157, please consider removing the sentence “However, a bayesian network meta-analysis allows multiple treatments to be simultaneously compared both direct and indirect evidence about therapeutic effects.” This sentence is better suited for the introduction or discussion sections.

Response: We had adjusted this sentence to the introduction.

16. In Table 1, the descriptions of 1, 2, 3 and 4, as presented in the outcome column, and which specific outcome they each refer to, should be presented in the note associated with the table.

Response: Explanations for 1, 2, 3 and 4 had been added in the Table footnotes.

17. In lines 223-224 please provide 95% confidence interval values for the change in outcome measures generated as a result of the sensitivity analysis.

Response: 95% confidence interval values had been added in lines 223-224 in “Robustness of conclusion” section.

18. Please clarify where the value of 32.72 in line 224 was obtained from. The value presented in Table 2 to this end appears to be 34.65 (95% CI 13.51 to 55.79).

Response: There was an error in recording, which had been corrected.

19. The second paragraph of the discussion section, specifically from line 247 to 266, requires significant improvement. The grammar and sentence structure makes the interpretation of results therein extremely confusing. This section should be revised and structured more carefully to prevent confusion.

Response: We had revised this section to correct incorrect grammar and sentence structure.

20. Lines 251 to 255 also appear to make baseless conclusions such as “Compared with pentoxifylline and berprost, cilostazol could increase by 8.76 and 4.15 meters, respectively”. As stated by the authors, the analysis to this end presented with statistical non-significance between outcomes, which directly contradicts this statement. While a hierarchy may be proposed on the basis of ranking probability, statements such as the above, made despite a clear indication of statistical non-significance in the outcome measure, are still considered erroneous.

Response: We had revised those sentences to correct inappropriate statements.

21. Please highlight the avenues for future research which have been opened up as a result of this study, or are needed to contextualize and further the findings in this study, more exhaustively. This could be in the form of studies evaluating more direct comparisons to complete more closed loops for future network meta-analyses to this end or studies investigating the efficacy of cilastozol plus beraprost in the walk distance outcome measures.

Response: In the last part of the discussion, we made some avenues for future research based on the findings.

---

## [Decision Letter · Decision Letter 1]

4 Sep 2022

PONE-D-22-10401R1Systematic review the efficacy and safety of cilostazol, pentoxifylline, berprost in the treatment of intermittent claudication: a network meta-analysisPLOS ONE

Dear Dr. liang,

Thank you for submitting your manuscript to PLOS ONE. After careful consideration, we feel that it has merit but does not fully meet PLOS ONE’s publication criteria as it currently stands. Therefore, we invite you to submit a revised version of the manuscript that addresses the points raised during the review process.

We look forward to receiving your revised manuscript.

Kind regards,

Tariq Jamal Siddiqi

Academic Editor

PLOS ONE

Journal Requirements:

Reviewers' comments:

Reviewer's Responses to Questions

**Comments to the Author**

1. If the authors have adequately addressed your comments raised in a previous round of review and you feel that this manuscript is now acceptable for publication, you may indicate that here to bypass the “Comments to the Author” section, enter your conflict of interest statement in the “Confidential to Editor” section, and submit your "Accept" recommendation.

Reviewer #2: (No Response)

2. Is the manuscript technically sound, and do the data support the conclusions?

Reviewer #2: Yes

3. Has the statistical analysis been performed appropriately and rigorously? 

Reviewer #2: Yes

4. Have the authors made all data underlying the findings in their manuscript fully available?

Reviewer #2: Yes

5. Is the manuscript presented in an intelligible fashion and written in standard English?

Reviewer #2: Yes

6. Review Comments to the Author

Reviewer #2: Liang et al. conducted a study on “Systematic review the efficacy and safety of cilostazol, pentoxifylline, beraprost in the treatment of intermittent claudication: a network meta-analysis”, in which they explored the efficacy and safety of cilostazol, pentoxifylline, and beraprost for the treatment of intermettient claudication as a result of lower extremity arterial occlusive disease. They found that cilostazol presented with the greatest improvement in both maximum and pain-free walk distance outcomes. A combination therapy of beraprost and cilostazol presented with the greatest improvement in ankle-brachial index outcome and of the active therapies, pentoxifylline, followed closely by cilostazol, presented with the lowest rates of adverse events. While the authors have done an excellent job of addressing most of the previously made comments, in my opinion, certain edits are still required:

1. While the authors have corrected the spelling of “beraprost” throughout the manuscript, they have not made the appropriate change in the title of the study. This would be an essential edit as it would play a significant role in the visibility of the article in searches.

2. In line 266, the phrase “leaving clinicians know about a drug to prescribe”, is poorly phrased and may confuse readers. The authors should revise it appropriately to improve readability.

3. In line 300, the authors should consider omitting the word “therefore” from the sentence “Therefore, in addition to exploring the direct....”.

7. PLOS authors have the option to publish the peer review history of their article (what does this mean?). If published, this will include your full peer review and any attached files.

Reviewer #2: **Yes: **Muhammad Talha Maniya

---

## [Author Response · Author response to Decision Letter 1]

5 Sep 2022

Dear reviewer,

Thank you very much for your comments and professional advice. These opinions help to improve academic rigor of our article. Based on your suggestion and request, we have made corrected modifications on the revised manuscript. 

Reviewer #2: 

1. While the authors have corrected the spelling of “beraprost” throughout the manuscript, they have not made the appropriate change in the title of the study. This would be an essential edit as it would play a significant role in the visibility of the article in searches.

Response: We have checked the manuscript and the title again, and corrected the misspelling.

2. In line 266, the phrase “leaving clinicians know about a drug to prescribe”, is poorly phrased and may confuse readers. The authors should revise it appropriately to improve readability.

Response: We have revised the phrase “leaving clinicians know about a drug to prescribe” as “providing guidance for clinicians in practice”.

3. In line 300, the authors should consider omitting the word “therefore” from the sentence “Therefore, in addition to exploring the direct....”.

Response: We have revised this sentence in line 300.

Thank you very much for your attention and time. Look forward to hearing from you.

---

## [Editor Report · Decision Letter 2]

15 Sep 2022

Systematic review the efficacy and safety of cilostazol, pentoxifylline, beraprost in the treatment of intermittent claudication: a network meta-analysis

PONE-D-22-10401R2

Dear Dr. liang,

We’re pleased to inform you that your manuscript has been judged scientifically suitable for publication and will be formally accepted for publication once it meets all outstanding technical requirements.

Kind regards,

Tariq Jamal Siddiqi

Academic Editor

PLOS ONE
---

## [Editor Report · Acceptance letter]

21 Oct 2022

PONE-D-22-10401R2 

Systematic review the efficacy and safety of cilostazol, pentoxifylline, beraprost in the treatment of intermittent claudication: a network meta-analysis 

Dear Dr. liang:

I'm pleased to inform you that your manuscript has been deemed suitable for publication in PLOS ONE. Congratulations! Your manuscript is now with our production department. 

Kind regards, 

on behalf of

Dr. Tariq Jamal Siddiqi 

Academic Editor

PLOS ONE